# Landscape Classification System Based on RKM Clustering for Soil Survey UAV Images–Case Study of the Small Hilly Areas in Jurong City

**DOI:** 10.3390/s22249895

**Published:** 2022-12-15

**Authors:** Zihan Fang, Wenhao Lu, Fubin Zhu, Changda Zhu, Zhaofu Li, Jianjun Pan

**Affiliations:** College of Resources and Environmental Sciences, Nanjing Agricultural University, Nanjing 210095, China

**Keywords:** landscape classification, land use, microtopography, soil survey, UAV, clustering

## Abstract

With the advantages of high accuracy, low cost, and flexibility, Unmanned Aerial Vehicle (UAV) images are now widely used in the fields of land survey, crop monitoring, and soil property prediction. Since the distribution of soil and landscape are closely related, this study makes use of the advantages of UAV images to classify the landscape to build a landscape classification system for soil investigation. Firstly, land use, object, and topographic factor were selected as landscape factors based on soil-forming factors. Then, based on multispectral images and Digital Elevation Models (DEM) acquired by UAV, object-oriented classification of different landscape factors was carried out. Additionally, we selected 432 sample data and validation data from the field survey. Finally, the landscape factor classification results were superimposed to obtain the landscape unit applicable to the system classification. The landscape classification system oriented to the soil survey was constructed by clustering 11,897 landscape units through the rough K-mean clustering algorithm. Compared to K-mean clustering, the rough K-mean clustering was better, with a Silhouette Coefficient of 0.26247 significantly higher than that of K-mean clustering. From the classification results, it can be found that the overall classification results are somewhat fragmented, but the landscape boundaries at the small area scale are consistent with the actual situation and the fragmented small spots are less. Comparing the small number of landscape boundaries obtained from the actual survey, we can find that the landscape boundaries in the landscape classification map are generally consistent with the actual landscape boundaries. In addition, through the analysis of two soil profile data within a landscape category, we found that the identified soil type of soil formation conditions and the landscape factor type of the landscape category is approximately the same. Therefore, this landscape classification system can be effectively used for soil surveys, and this landscape classification system is important for soil surveys to carry out the selection of survey routes, the setting of profile points, and the determination of soil boundaries.

## 1. Introduction

Soil survey is a basic method for the field study of soils, through which we can understand the classification and distribution of soils. During soil surveys in the field, we need to dig up the soil and analyze and compare the soil profile pattern and the surrounding environment, which often requires many resources to excavate the soil profile, so we need a more convenient method to conduct soil surveys. Landscape and soil are correlated [1], and Nikiforova [2] proposed to create a system of classification of natural soil landscapes, considering natural soil is a self-sufficient system, but also as a subset of natural landscape systems, i.e., soil as a derived landscape element. We can consider that the boundary of soil and the boundary of landscape overlap under certain circumstances, so a classification of the landscape under certain conditions also implies a classification of soil. Nowadays, studies have been conducted to systematically classify soils at the soil series level through landscape variability [3,4,5,6,7], and relevant technical guidelines for soil surveys have been published [8] to guide practical soil surveys.

In recent years, Unmanned Aerial Vehicle (UAV) technology has developed rapidly, has the advantage of the flexibility and low cost, and has been widely used in soil and environmental resource investigation. Compared with satellite images, which have the disadvantage of errors due to cloud cover and atmospheric reflection, UAV images have advantages. A large number of studies [9,10] have been conducted using UAVs as piggyback platforms to predict soil properties with generally high prediction accuracy. On the one hand, traditional soil surveys are labor-intensive, while UAVs have lower costs compared to traditional soil surveys, and on the other hand, UAVs serve as a good piggyback platform, and the advanced sensors on board UAVs can be used to obtain multi-dimensional data that cannot be obtained by traditional soil surveys. Ma [11] achieved the identification of *Seriphidium transiliense*, *Ceratocarpus arenarius*, and bare ground by Red, Red Edge, and Near InfraRed (NIR) bands obtained from UAV imagery. Yan [12] used high-resolution UAV imagery for vegetation cover prediction. These predicted or identified indicators are within the scope of soil surveys, so it is necessary to introduce UAVs into soil surveys. In addition, the results of these studies are all highly accurate, which provides a basis for high-precision soil classification.

Some studies have used UAVs for studies such as landscape pattern analysis [13,14] and even landscape classification [15], but in these studies only the distribution of the landscape pattern was considered, focusing purely on a single attribute of topography or ground cover as the main component of the landscape. Therefore, these studies may only be applied to national land surveys but not to soil classification. In a landscape classification system oriented to soil classification, it is important to include not only factors that facilitate the differentiation of landscapes, but also factors that affect soil occurrence. There are many approaches to landscape classification, including decision trees for unsupervised classification [3,4,5], convolutional neural networks for supervised classification [16,17], Landscape Character Assessment (LCA) [18], and Parameter Configuration [19,20], all of which can be used for landscape classification. However, most of these methods rely heavily on personal experience in understanding and evaluating the landscape, and therefore cannot provide a quantitative description of the composition of the landscape. Some metrics can be used to describe the landscape by quantification. Harrison [19] proposes the use of geomorphic, a physical landform classification scheme, to visualize and characterize the parameter configuration landscape. However, these parameters can only be used to reflect differences between landscapes and cannot describe the composition of the landscape, which is unsuitable for soil surveys.

Therefore, this study aims to classify landscapes based on high-precision images from UAVs. The quantifiable components of the soil landscape (i.e., landscape factors) are selected and classified based on soil-forming factors and related soil classification studies. The landscape factors are then superimposed to obtain landscape units. Finally, all landscape units are clustered to obtain a soil survey-oriented landscape classification system. Based on the correlation between soil and landscape, the final results can provide a reference for determining soil types and boundaries, and also provide an important basis for route selection and profile point setting for field soil surveys.

## 2. Materials and Methods

### 2.1. Study Area

The study area is located in the Longshan Reservoir catchment and is northeast and southwest of Jurong City, Jiangsu Province, People’s Republic of China, with an area of 9.02 square kilometers, including 7.46 square kilometers of land area. The study area is in latitude 32°2′30.66″ N to 32°6′10.08″ N and longitude 119°13′29.55″ E to 119°16′50.69″ E (Figure 1). It has a north subtropical monsoon climate. Rainfall is abundant, with an annual precipitation of 1088.2 mm, and the annual average dryness is around 1.0. The average annual temperature is 15.6 °C, the average daily temperature above 10 °C for the crop growing period is 226 days on average, the frost-free period is 229 days, and the crop maturity system is biannual [21]. Crops are mainly rice, wheat, and corn. The southwest of the study area is plain, with absolute elevation ranging from 20 m to 70 m, and the plain area gradually increases in elevation from southeast to northwest. The northeast of the study area is hilly, and its easternmost part is GaoLi Mountain, with absolute elevation ranging from 50 m to 130 m. The topography of the study area is complex, with a large number of hills and terraces, and the development of gullies and valleys is between the hills and terraces with drastic elevation changes. The soil-forming parent material in the study area is mainly the lower Shu loess parent material, rocky weathering slope accumulation parent material, and alluvial parent material. The slopes and foothills below the middle of the stony hills are mostly rocky weathered slope deposit parent material, and the slopes are mostly in the range of 6° to 15°. The parent material type of the muddy hills, the hillocks, and the gully land is mainly XiaShu loess with deep loess layer. The parent material type of terraces is mainly alluvial parent material [3].

### 2.2. Data Collection and Data Pre-Processing

The experiment uses the DJI Phantom 4 Multispectral Edition as the flight platform. The UAV is equipped with an all-in-one spectral imaging system that integrates six cameras: visible, red, blue, green, red edge, and near infrared, which are responsible for visible and multispectral imaging, respectively. The UAV is also equipped with Real-Time Kinematic (RTK) and TimeSync time synchronization systems, which can achieve centimeter-level positioning. Among them, the camera resolution is about 2.12 million pixels, the focal length is 5.74 mm, and the spectral parameters are shown in Table 1.

This study was conducted between late December 2021 and early January 2022, with clear and cloudless weather from 10:00 to 14:00 on the same day. The flight altitude was 120 m, the heading overlap rate was 80%, the side overlap rate was 60%, the whole flight was automatic, and the whiteboard was taken between flights to obtain the corrected images from the UAV multispectral camera. At the same time, sample points were collected on the same day, the landscape of the sample points was observed and photographed, the coordinates of the sample points were located using Global Positioning System (GPS), and the information and coordinates of the sample points such as land use type, object type, and microtopography were recorded. A total of 432 sample points were collected, as shown in Figure 1, including 308 training samples. Twelve soil profile sample sites were collected between July 2022 and August 2022. The morphology of the soil profile was observed, and the soil genetic horizons were classified according to differences in properties such as compactness, color, and redox degree of the soil, while the soil-forming environment, GPS, and the morphological characteristics of soil genetic horizon were recorded. Morphological characteristics of the soil genetic horizon include indicators of genetic horizon thickness, boundary, color, root, texture, structure, nascent body, invasive body, etc.

DJI Terra software was used for stitching and radiometric correction of the UAV images, and five single-band images of blue, green, red, red edge, near infrared, and Digital Elevation Model (DEM) were obtained. In ENVI5.3, the five single-band images were band-synthesized and resampled to 0.1 m, and the DEM was also resampled to 0.1 m.

### 2.3. Selection and Classification of Landscape Factors

According to previous studies, landscape factors oriented to soil classification should include land use type, object type, and topography [3,4,5]. In Simensen’s [22] study on the correlation of landscape factors, topographies, ground cover, vegetation, topography, and soil were ranked in the top correlations, respectively. Some landscape classifications combine landforms/surface relief with land use types to classify landscapes [23,24]. In many bottom-up constructed landscape classification systems, the lowest landscape unit is composed of an overlay of topography, soil genotype, major vegetation community complex, and current land cover type [24]. In this study, three categories of land use type, object type, and topography were selected as landscape factors for this landscape classification system by combining the previous research results and research objectives.

The UAV hyperspectral images were subjected to the object-oriented classification of land use types and object types in eCognition Developer 9.0. Image segmentation was performed using multilevel segmentation, and the Estimation of Scale Parameter (ESP) scale evaluation tool was used to obtain the best segmentation effect scale for features [25], and the classifier algorithm was a random forest algorithm, and feature parameters were selected for land use type classification and object type classification concerning other studies [25,26,27,28]: the mean and standard deviation value of each band, normalized difference vegetation Index (NDVI), normalized difference water index (NDWI), red edge normalized difference vegetation index (REDNDVI), Length/Width, shape index, and gray level co-occurrence matrix (GLCM) to calculate the texture features [29], and the specific calculation formula are shown in Table 2.

Topography classification using the Geomorphons (GM) topography classification method [30], with the search radius of the search window set to 20 [31] and the flatness threshold set to 5° [32], can achieve better classification results. The lower search radius reflects the terrain at a small scale, i.e., microtopography. The slope reference includes flat (≤2°), gentle slope (2 to 6°), medium gentle slope (6° to 15°), medium slope (15° to 25°), and steep slope (≥25°) [32]. The slope direction is divided into 4 categories: north, east, south, and west. The relative elevation is divided into nine categories from 30 m to 100 m in a gradient of 10 m, plus 0–30 m and 100–130 m. The details are shown in Table 3 and Table 4.

### 2.4. Rough K-Means Clustering

Lingras [33,34] proposed the RKM algorithm, a clustering inspired by intervals that exploit the basic properties of the original rough set theory, namely the concepts of lower and upper approximation. The method treats each cluster as a rough set. Objects that can belong to a cluster with certainty are divided into the lower approximation set of that class of clusters, while data samples with uncertain affiliation are divided into the upper approximation set of multiple clusters. Thus, an object can belong to more than one cluster, and the objects divided into upper approximation sets can be regarded as “buffers” between different clusters. The clustering process is as follows.

Step 1: For each cluster and object, find the distance d and the threshold T.

Step 2: Use the classification criteria (R1 and R2) to classify the data objects into lower and higher estimates.

Step 3: Compute the new cluster centers according to the following expressions.
(1)$$z_{i}=\{\begin{array}{c} \frac{\sum_{v\in \underline{A}\left(c_{i}\right)}v}{\left|\underline{A}\left(c_{i}\right)\right|}\;if\left[\underline{A}\left(c_{i}\right)\neq \emptyset \;and\;\bar{A}\left(c_{i}\right)-\underline{A}\left(c_{i}\right)=\emptyset \right]\\ \frac{\sum_{v\in \bar{A}\left(c_{i}\right)-\underline{A}\left(c_{i}\right)}v}{\left|\bar{A}\left(c_{i}\right)-\underline{A}\left(c_{i}\right)\right|}\;if\left[\underline{A}\left(c_{i}\right)=\emptyset \;and\;\bar{A}\left(c_{i}\right)-\underline{A}\left(c_{i}\right)\neq \emptyset \right]\\ w_{l}\times \frac{\sum_{v\in \underline{A}\left(c_{i}\right)}v}{\left|\underline{A}\left(c_{i}\right)\right|}+w_{u}\times \frac{\sum_{v\in \bar{A}\left(c_{i}\right)-\underline{A}\left(c_{i}\right)}v}{\left|\bar{A}\left(c_{i}\right)-\underline{A}\left(c_{i}\right)\right|}\;if\left[\underline{A}\left(c_{i}\right)\neq \emptyset \;and\;\bar{A}\left(c_{i}\right)-\underline{A}\left(c_{i}\right)\neq \emptyset \right] \end{array}$$
$w_{l}+w_{u}=0\;\mathrm{and}\;w_{l}>w_{u,\;}w_{l}$ and $w_{u}$ correspond to the relative importance of the lower and upper approximations, respectively.

Step 4: Repeat steps Step 2 to Step 4 until the algorithm converges.

### 2.5. Kappa Coefficient

The Kappa coefficient is a measure of classification accuracy and was proposed by Cohan [35]. Its calculation formula is as follows.
(2)$$\mathrm{Kappa}=\frac{P_{0}-P_{e}}{1-P_{e}}$$
(3)$$P_{0}=\frac{\sum_{i=1}^{n}P_{ii}}{N}$$
$P_{0}$ is the overall accuracy of the classification, $P_{e}$ expresses the probability that the classification result due to chance is consistent with the type of ground survey data, n is the number of types classified, N is the total number of samples, and $P_{ii}$ is the number of samples of type i that is correctly classified.

When the Kappa coefficient is closer to 1, it means that the classification effect is better, and the classification result is more consistent with the actual situation.

### 2.6. Silhouette Coefficient

Silhouette Coefficient (SC) for cluster validity analysis clearly shows not only the high similarity of sample points within clusters but also the high object dissimilarity between clusters [36]. The silhouette coefficient of object i is calculated as follows:(4)$$s\left(i\right)=\frac{b\left(i\right)-a\left(i\right)}{max\left\{a\left(i\right),b\left(i\right)\right\}}$$
a(i) is the intra-cluster dissimilarity, the average distance from object i to other objects in the same cluster, and b(i) is the inter-cluster dissimilarity, the average distance from object i to all objects in some other cluster C.

If s(i) is close to 1, it means that object i is more reasonably clustered; if s(i) is close to −1, it means that object i is more deserving of classification into a certain cluster C; if s(i) is close to 0, it means that object i is on the boundary of two clusters. The Mean of the Silhouette Coefficient (MSC) can be expressed as:(5)$$\bar{S}=\frac{1}{n}\sum_{i=1}^{n}s_{i}$$

### 2.7. Determination of the Optimal Number of Clusters

The RKM clustering algorithm still clusters the dataset with a determined number of classes k and randomly selected initial clustering centers. However, usually, the clustering number k cannot be determined in advance, and the randomly selected initial clustering centers tend to make the clustering results unstable [37]. Therefore, the optimal number of clusters k needs to be determined. The algorithm for determining the optimal number of clusters in this paper is summarized as follows.

Step 1: Select the search radius [$k_{min}$, $k_{max}$] of the clustering number. $k_{min}$ = 20 and $k_{max}$ = 100 are taken in this paper according to the experimental purpose.

Step 2: Take all the numbers in the search radius [$k_{min}$, $k_{max}$] for the clustering number k respectively

Step 3: Perform RKM clustering.

Step 4: Count the clustering results and compare the mean of Silhouette Coefficients $\bar{S}$ of different clustering numbers k.

Step 5: Output the best clustering number k.

## 3. Results and Analysis

### 3.1. Classification and Evaluation of Landscape Factors

Based on texture, geometric features, and spectral features for land use types and object types, the land use types and object types obtained from the field survey are used as training samples for object-oriented classification to obtain land use type maps and object type maps. The topography was classified into 9 categories by the GM topography classification method, and topographic classification maps were generated by DEM. Reclassify the DEM to get the DEM classification map. Reclassify the slope map and aspect map generated by DEM to get the slope classification map and aspect map.

The classification accuracy of land use type and object type as landscape factors significantly affect the classification accuracy of the landscape, and the confusion matrix accuracy evaluation and statistics were conducted on Arcmap10.8 based on the field survey sample points and classification results. Among them, the total classification accuracy of land use types was 93.38% with a Kappa coefficient of 0.9216, the total classification accuracy of object types was 91.18% with a Kappa coefficient of 0.8999, and the production accuracy and user accuracy of land use types and object types were mostly above 90%.

From the classification results, it can be seen that the main land use type in the study area is arable land, with 16.60% and 15.69% of the whole study area in paddy fields and dry land, respectively, and the main crops are rice, wheat, edible rape, and corn, which are consistent with the actual situation. In addition, a small amount of finely watered land exists near the greenhouses and construction sites for growing vegetables. According to the field conditions, most of the crops in the study area are planted twice a year, with winter wheat and rice grown in paddy fields, rape and wheat grown in drylands, and a significant portion of the drylands are planted with corn starting in summer. In addition to arable land, a large number of orchards, tea gardens, nurseries, and other gardens are used to grow cash crops in the study area, accounting for 13.53% of the total study area. Natural forests in the hills of the northeastern part of the study area and small patches of plantation forests planted in the central and southeastern plains of the study area make up the entire study area, accounting for 17.15% of the total study area. The wasteland consisted mainly of river floodplains, reedbeds near the watershed, and a small amount of long-abandoned arable land, covering 9.44% of the total study area. Finally, the remaining building land and water area accounted for 6.61% and 19.87% of the total study area.

In this study, it is possible to identify and classify oilseed rape finely distributed on the roadside or fieldside and watered land distributed near houses, which is important for the classification of fragmented landscapes. For example, some of the paddy fields in the study area are flooded in winter, and the water left on the surface of the paddy fields may be misclassified as water, and there may be a strip of water like a ridge in the paddy fields, so there is a need to merge the land use types and object types appropriately.

### 3.2. Landscape Classification

Since the purpose of this study is a landscape classification system oriented to soil survey, landscape differences of building sites and waters were not considered in the landscape classification, and all landscape factors were superimposed after removing building sites and waters for a total of 11,897 landscape units.

The optimal number of clusters was calculated to be 95, and the mean of Silhouette Coefficient $\bar{S\;}\;$= 0.26247. The landscape factors were clustered by the RKM algorithm, the results were shown in Figure 2, and the dominant landscape unit in the category was named the landscape category.The naming format was “topography–slope–slope direction–elevation–land use type—object type”, and water and building land are divided into 97 categories of landscape, as shown in Table 5.

The landscape classification results were analyzed for t9-s1-a4-d4-l7-o10, a landscape category with an area of 1468.64 ha, accounting for 1.63% of the total study area. It consists of 61 landscape units clustered together, mainly distributed in the plain in the southwest of the study area. The relative elevation of this landscape category ranges from 30 m to 50 m, the slope is between 0° and 6°, and the slope direction is mainly north and east. The main land use type is paddy field, and the object type is mainly paddy (Figure 3). Based on the two soil profile sample sites from the field soil survey (Figure 4), the soils of this landscape type can be identified to the subgroup level under the guidance of the Chinese Soil Taxonomy (CST) criteria. The names in order from soil order to subgroup are Anthrosols, Stagnic Anthrosols, Fe-Stagnic Anthrosols, and Ordinary Fe-Stagnic Anthrosols.

According to the profile record sheet, anthrostagnic epipedon and hydragric horizon were present at the profile sampling sites. Anthrostagnic epipedon was mainly caused by long-term tillage, and hydragric horizon was mainly caused by changes in soil groundwater level [38]. Its hydragric horizon also had a large number of Fe-Mn nodules and no gleyic features, indicating that the cultivated land was well irrigated, the soil was well drained, and the soil development was good. These diagnostic horizons and diagnostic features are mainly influenced by anthropogenic tillage and water table, which are inseparable from land use and topography. t9-s1-a4-d4-l7-o10 has clear boundaries and concentrated distribution, and when comparing the influence of each landscape factor on clustering, we found that land use and topography in this landscape category influence land use, and topography on the clustering was found to be dominant in this landscape category and had a large influence on the clustering results.

However, the dominant landscape factors are not the same among different landscape categories, for example, topography has the greatest influence on clustering in class t6-s3-a1-d5-l7-o8, and topography and land use jointly influence the clustering results in t9-s2-a3-d5-l3-o3. Therefore, the same landscape factor may have different effects on the clustering of different landscape types. Additionally, in the actual soil classification, paddy fields, as one of the land types most severely affected by man, have huge differences in physical and chemical properties from other soils due to long-term tillage and irrigation. Land use and crop type are often decisive in the classification of soils related to paddy fields, so the use of land as the main landscape factor affecting landscape classification in the t9-s2-a2-d5-l1-o1 landscape category is very consistent with the actual soil classification.

### 3.3. Landscape Outcome Assessment

In this study, 11,897 landscape units were clustered by RKM clustering, plus water and building sites were divided into 97 classes with the MSC $\bar{S\;}\;$= 0.26247. In addition, K-mean clustering was set as the control group, and the mean of the Silhouette Coefficient was calculated for the number of clusters from 20 to 100, and the results are shown in Figure 5.

The MSC of RKM clustering is higher than the MSC of K-mean clustering 0.16436 when the number of clusters is equal to the optimal number of clusters 95. In addition, the MSC of RKM clustering is higher than the highest MSC of K-mean clustering 0.16806 in the whole range. Therefore, the clustering effect of rough K-mean clustering is significantly higher than that of K-mean clustering.

From the final classification results, although the overall classification results are somewhat fragmented, the landscape boundaries at the small area scale are in line with the actual situation and the fragmented small spots are less. By comparing a small number of landscape boundaries obtained from the actual survey, the landscape boundaries in the landscape classification map are generally consistent with the actual landscape boundaries.

## 4. Discussion

Landscape boundaries and soil boundaries tend to overlap in general, and setting several inspection profiles within a particular landscape to understand soil variation within a landscape category can determine that the same soil is often present in the same landscape [39]. Therefore, in the actual soil survey, after classifying the landscape by UAV images or satellite images, a reasonable soil survey route is set based on the landscape classification map. Each route should contain as many landscape types as possible, and a small number of soil and check profiles should be taken in each landscape type [40]. If the soil types within the same landscape type are the same, it can be assumed that the boundary of the landscape is the boundary of the soil. Therefore, landscape classification is of great importance for soil investigation, and it has great advantages to obtain data in some inaccessible areas by UAV. Compared with remote sensing images, which are easily and costly affected by cloud cover and atmosphere, UAV images are not only more accurate and less affected by cloud cover and atmosphere, but also have greater advantages in terms of convenience and low cost of acquiring data [41].

During the actual soil survey, it can be found that soil changes are gradual and therefore the boundaries of landscape categories may be blurred. In some classifications, some objects are in between two different categories, and thus these objects cannot be classified effectively. Therefore, many classifications reflect the “fuzzy relationship” between types through the affiliation of fuzzy clusters [42,43]. However, this affiliation relationship cannot be shown on the graph in practical applications. In this study, rough sets are introduced into the landscape classification through rough clustering, where a landscape unit may belong to more than one landscape category. These landscape units can then be regarded as buffers between different landscape categories, i.e., asymptotic processes of soil properties in natural situations.

## 5. Conclusions

In this study, landscape classification in the Jurong district is achieved by RKM clustering with the advantage of convenience and low cost of high-precision UAV data. The results are in line with the actual situation and the experiment obtained the expected results. Compared with the K-mean clustering results, the clustering effect is better, which shows the superiority of UAV data and the RKM clustering algorithm. It provides a basis for soil survey and soil classification and further provides a reference meaning for landscape classification that can be widely used.

## Figures and Tables

**Figure 1 sensors-22-09895-f001:**
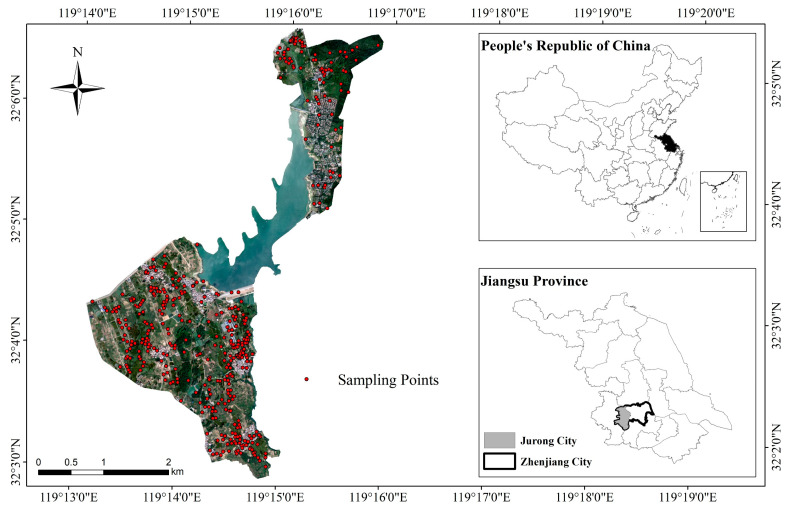
Map of the study area and sampling points distribution.

**Figure 2 sensors-22-09895-f002:**
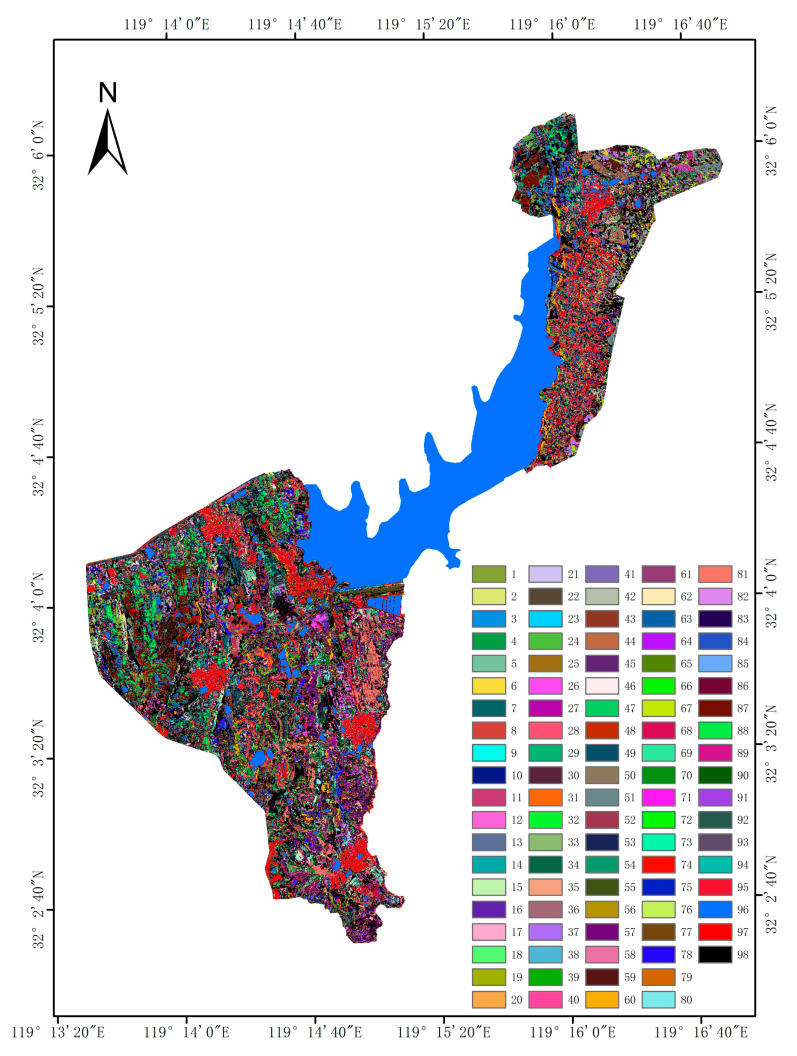
Map of landscape classification.

**Figure 3 sensors-22-09895-f003:**
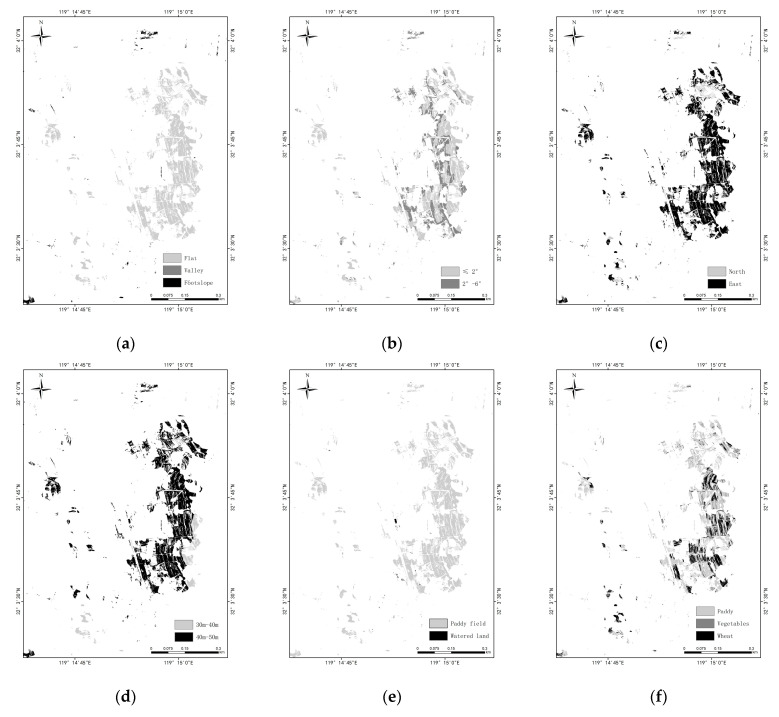
Landscape factor composition of t9-s1-a4-d4-l7-o10 (**a**) topography (**b**) slope (**c**) aspect (**d**) dem (**e**) land use (**f**) object.

**Figure 4 sensors-22-09895-f004:**
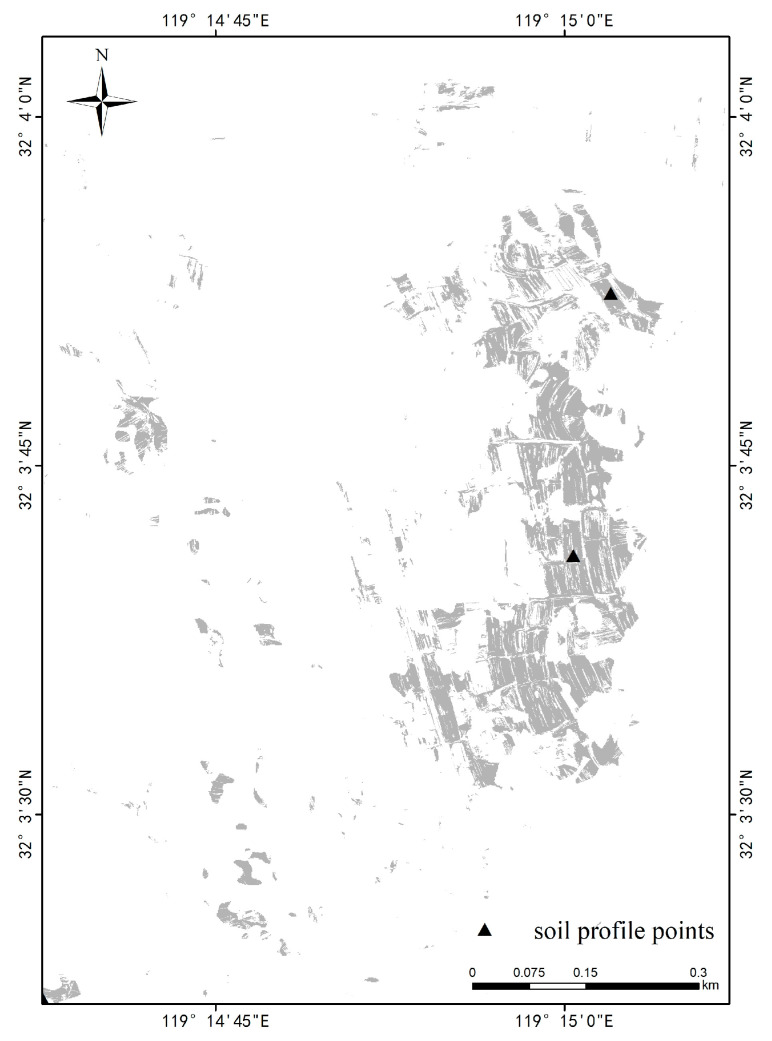
Section sample point diagram.

**Figure 5 sensors-22-09895-f005:**
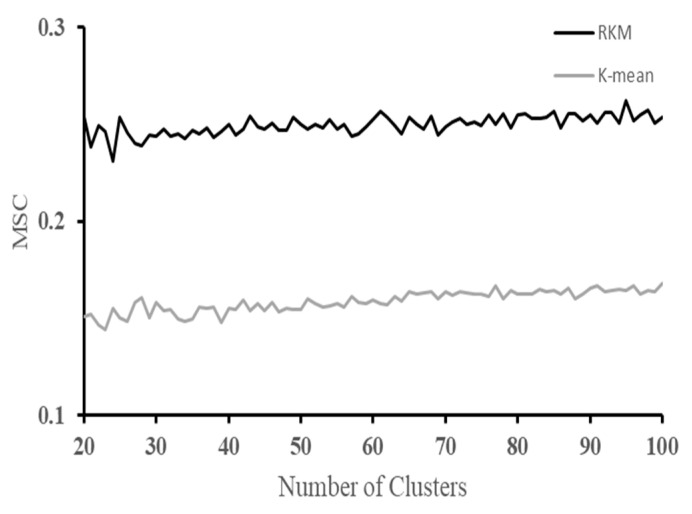
Chart of the Mean of Silhouette Coefficient of RKM and K-mean.

**Table 1 sensors-22-09895-t001:** Spectral parameters of DJI Phantom 4 Multispectral camera.

Serial Number of Spectrum	Name of Spectrum	Central Wavelength/nm	Full Width at Half Peak/nm
$B_{1}$	Blue	450	32
$B_{2}$	Green	560	32
$B_{3}$	Red	650	32
$B_{4}$	Red Edge	730	32
$B_{5}$	Near InfraRed	840	52

**Table 2 sensors-22-09895-t002:** Calculation formula of feature parameter.

Feature Category	Feature Parameter	Formula
Spectrum feature	Mean	$\bar{B_{k}}=\frac{1}{n}\sum_{i=1}^{n}B_{ki}$
	Standard Deviation	$\sigma_{L}=\sqrt{\frac{1}{n}\sum_{i=1}^{n}{\left(B_{ki}-\bar{B_{k}}\right)}^{2}}$
Shape feature	NDVI	$\mathrm{NDVI}=\frac{B_{5}-B_{3}}{B_{5}+B_{3}}$
	NDWI	$\mathrm{NDWI}=\frac{B_{2}-B_{5}}{B_{2}+B_{5}}$
	REDNDVI	$\mathrm{REDNDVI}=\frac{B_{5}-B_{4}}{B_{5}+B_{4}}$
Geometric feature	Length/Width	$\gamma =\frac{l}{w}=\frac{eig_{1}\left(S\right)}{eig_{2}\left(S\right)}$
	Shape Index	$I=\frac{L}{4\sqrt{A}}$
Texture features	GLCM	P(i,j,d,θ)=#{(k,l),(m,n)∈(M×N)|f(k,l)=i,f(m,n)=j}

Notes: k is the band, n is the number of pixels contained in the image object, $B_{ki}$ is the pixel value,$\;\bar{B_{k}}$ is the mean value of the band $L_{i}$, l is the length of the outer wrapping rectangle of the object; w is the width of the outer wrapping rectangle of the object, L is the length of the boundary of the image object, A is the area, P(i,j,d,θ) is the pixel from the point (k,l) with grayness i in the image, and the statistics with its distance d=(m−k,n−1) of the point (m,n) with grayscale j at the same time, d is the relative distance expressed in terms of the number of pixels, and θ is the relative direction.

**Table 3 sensors-22-09895-t003:** Categories of terrain, slope, and aspect.

Topography		Slope			Aspect	
Code	Category	Code	Category	Slope	Code	Category
1	Flat	1	Flat	≤2°	1	North
2	Ridge	2	Gentle Slope	2°–6°	2	East
3	Spur	3	Gentle–Middle Slope	6°–15°	3	South
4	Hollow	4	Middle Slope	15°∼25°	4	West
5	Valley	5	Steep Slope	≥25°		
6	Peak					
7	Shoulder					
8	Slope					
9	Footslope					
10	Pit					

**Table 4 sensors-22-09895-t004:** Categories of elevation, land use, and object.

Elevation		Land use		Object	
Code	Elevation	Code	Category	Code	Category
1	0–30 m	1	Paddy field	1	Paddy
2	30–40 m	2	Watered land	2	Vegetables
3	40–50 m	3	Dry land	3	Wheat
4	50–60 m	4	Orchard	4	Rape
5	60–70 m	5	Tea plantation	5	Maize
6	70–80 m	6	Nursery	6	Sesame
7	80–90 m	7	Woodland	7	Orchard
8	90–100 m	8	Wasteland	8	Tea
9	100–130 m	9	Greenhouse	9	Nursery
		10	Building	10	Evergreen forest
		11	Water	11	Mixed forests
				12	Deciduous forest
				13	Wasteland
				14	Building
				15	Water

**Table 5 sensors-22-09895-t005:** Category of landscape factor.

Landscape	Area	Area Percentage	Code	Landscape	Area	Area Percentage	Code
t1-s1-a1-d4-l3-o3	1426.999	1.58%	1	t6-s2-a4-d9-l3-o3	25.5882	0.03%	50
t2-s1-a1-d5-l3-o3	324.39	0.36%	2	t6-s2-a4-d9-l7-o10	812.6314	0.90%	51
t2-s2-a3-d5-l1-o3	71.4506	0.08%	3	t6-s3-a1-d5-l7-o8	2000.857	2.22%	52
t3-s1-a1-d3-l1-o1	979.5747	1.09%	4	t6-s3-a1-d5-l9-o2	168.0305	0.19%	53
t3-s1-a1-d5-l3-o5	863.9857	0.96%	5	t6-s3-a1-d9-l1-o1	62.9914	0.07%	54
t3-s1-a2-d7-l7-o10	129.5776	0.14%	6	t6-s3-a4-d3-l3-o3	820.5404	0.91%	55
t3-s1-a3-d5-l3-o3	1008.926	1.12%	7	t6-s4-a1-d2-l1-o1	389.4215	0.43%	56
t3-s1-a3-d7-l3-o5	15.9806	0.02%	8	t6-s4-a2-d2-l4-o7	607.1563	0.67%	57
t3-s1-a4-d5-l9-o2	49.6719	0.06%	9	t6-s4-a2-d5-l3-o3	648.872	0.72%	58
t3-s2-a1-d6-l1-o1	192.4146	0.21%	10	t6-s4-a2-d7-l7-o12	563.2972	0.62%	59
t3-s2-a4-d5-l1-o1	361.8445	0.40%	11	t6-s4-a4-d4-l7-o13	1315.331	1.46%	60
t3-s3-a1-d9-l7-o12	140.2821	0.16%	12	t6-s4-a4-d5-l1-o1	344.0997	0.38%	61
t3-s3-a4-d4-l7-o10	972.6522	1.08%	13	t6-s4-a4-d8-l3-o5	162.8713	0.18%	62
t3-s3-a4-d9-l3-o5	39.3105	0.04%	14	t7-s1-a4-d3-l3-o3	207.3702	0.23%	63
t4-s1-a1-d2-l3-o3	469.9133	0.52%	15	t7-s2-a1-d2-l7-o9	350.6233	0.39%	64
t4-s1-a4-d2-l7-o8	471.4972	0.52%	16	t7-s2-a3-d2-l1-o3	43.0202	0.05%	65
t4-s1-a4-d6-l7-o10	376.4227	0.42%	17	t7-s2-a3-d9-l1-o1	24.0997	0.03%	66
t4-s2-a1-d5-l1-o3	93.5883	0.10%	18	t7-s2-a4-d9-l7-o10	330.6616	0.37%	67
t5-s1-a3-d6-l3-o5	119.127	0.13%	19	t7-s3-a1-d5-l7-o10	544.4619	0.60%	68
t5-s1-a4-d3-l1-o3	63.526	0.07%	20	t7-s3-a1-d7-l3-o5	49.7057	0.06%	69
t5-s1-a4-d5-l7-o12	277.9403	0.31%	21	t7-s3-a4-d9-l9-o2	50.7333	0.06%	70
t5-s2-a1-d3-l3-o5	381.3906	0.42%	22	t7-s4-a1-d2-l1-o1	103.57	0.11%	71
t5-s2-a1-d5-l1-o3	127.9758	0.14%	23	t7-s4-a1-d4-l3-o3	182.2174	0.20%	72
t5-s2-a1-d5-l9-o2	77.3689	0.09%	24	t7-s4-a2-d3-l7-o13	121.2364	0.13%	73
t5-s2-a1-d7-l7-o12	198.5637	0.22%	25	t7-s4-a3-d6-l3-o3	168.7981	0.19%	74
t5-s2-a4-d2-l3-o5	772.3075	0.86%	26	t7-s4-a3-d6-l7-o10	414.9493	0.46%	75
t5-s2-a4-d2-l8-o13	264.2767	0.29%	27	t7-s4-a4-d4-l1-o2	194.7145	0.22%	76
t5-s2-a4-d5-l1-o1	107.1413	0.12%	28	t7-s4-a4-d4-l3-o7	61.2594	0.07%	77
t5-s2-a4-d9-l1-o1	31.4418	0.03%	29	t8-s1-a1-d4-l7-o8	1251.572	1.39%	78
t5-s2-a4-d9-l7-o11	115.1422	0.13%	30	t8-s2-a3-d2-l8-o13	517.4768	0.57%	79
t5-s3-a1-d5-l7-o10	577.5642	0.64%	31	t8-s2-a4-d2-l3-o5	567.8578	0.63%	80
t5-s3-a1-d9-l3-o3	40.225	0.04%	32	t9-s1-a1-d2-l1-o3	1216.079	1.35%	81
t5-s3-a4-d8-l3-o5	82.1636	0.09%	33	t9-s1-a1-d6-l7-o13	299.9196	0.33%	82
t5-s3-a4-d8-l9-o2	21.2627	0.02%	34	t9-s1-a4-d3-l1-o3	380.9284	0.42%	83
t5-s4-a1-d2-l7-o10	289.9934	0.32%	35	t9-s1-a4-d4-l7-o10	1468.644	1.63%	84
t5-s4-a2-d3-l1-o1	79.8899	0.09%	36	t9-s1-a4-d9-l3-o5	27.4092	0.03%	85
t5-s4-a3-d8-l7-o12	247.3944	0.27%	37	t9-s2-a1-d2-l7-o12	429.399	0.48%	86
t5-s4-a4-d4-l3-o3	187.5225	0.21%	38	t9-s2-a2-d5-l1-o1	2456.642	2.72%	87
t5-s4-a4-d4-l7-o9	783.618	0.87%	39	t9-s2-a3-d5-l3-o3	1827.01	2.03%	88
t6-s1-a1-d3-l3-o3	377.7436	0.42%	40	t9-s2-a3-d8-l7-o11	139.7857	0.15%	89
t6-s1-a1-d9-l7-o10	142.0589	0.16%	41	t9-s2-a4-d2-l1-o1	832.891	0.92%	90
t6-s1-a4-d3-l1-o1	862.9392	0.96%	42	t9-s3-a1-d7-l3-o5	21.2196	0.02%	91
t6-s1-a4-d3-l7-o9	765.05	0.85%	43	t9-s3-a2-d3-l3-o5	54.4972	0.06%	92
t6-s1-a4-d6-l7-o13	1480.8	1.64%	44	t9-s3-a4-d5-l3-o5	64.5773	0.07%	93
t6-s2-a1-d3-l7-o13	1339.79	1.49%	45	t9-s4-a2-d4-l7-o13	107.5602	0.12%	94
t6-s2-a1-d5-l1-o1	508.4847	0.56%	46	t10-s1-a1-d4-l7-o9	510.7356	0.57%	95
t6-s2-a1-d5-l7-o8	680.4379	0.75%	47	Water	17925.71	19.87%	96
t6-s2-a1-d9-l3-o5	51.557	0.06%	48	Building	5963.977	6.61%	97
t6-s2-a4-d5-l3-o3	1297.76	1.44%	49	Buffer Zone	23035.35	25.54%	98

## Data Availability

Not applicable.

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
