# Peer review of "Landscape Classification System Based on RKM Clustering for Soil Survey UAV Images–Case Study of the Small Hilly Areas in Jurong City"

_sensors, 2022, doi:10.3390/s22249895_

Round 1

Reviewer 1 Report

In the PDF you will find corrections.
Generally speaking, some sentences are too long and the whole text needs additional proofreading.

Author Response

The following changes have been made in response to your comments and suggestions.
(1) Revise some sentences that are too long.
(2) Some incorrect uses of proper nouns were corrected.
(3) Some incorrect formatting was corrected.

Thank you very much for your valuable suggestions!

Reviewer 2 Report

This paper is interesting and takes important problem of landscape and soil survey with the UAV techniques. Paper structure and source materials are appropriate. The proposed methodology and obtained results are encouraging and allows for future soil survey in the small areas. Below I mention some problems with your article.

1. Title needs correction. I propose: ‘Landscape classification system based on RKM clustering for soil survey UAV images - case study of the small hilly areas in Jurong City’.

2. Abstract need correction. I propose re-writing this text and order the information about your investigation, research aims, methods and results.

3. The Introduction section need correction. Please indicate more clarity reasons of your investigation and aims of this study. I propose limited this chapter to description of use the UAV methods to the solution of landscape classification problem.

4. Material and Methods.

Study area section needs supplementation. I propose to add one paragraph with more detailed description of study area, e.g. geological units, landforms and hydrography, soil cover and natural vegetation. Add some information about land cover and land use (with settlement pattern).

3. Results and Analysis.

Can you explain, what is mean the ‘landscape phases’?

4. Conclusions and Discussion need correction. In my opinion you should discuss your obtained results with soil type in the sampling points (test area) or comparing with references.  

 Detailed comments are provided in the text (enclosed pdf).

 To discussion

1/ In my opinion the landscape or landscape unit it is a big area – above 1 km2. In your research the landscape class overlap with object.

2/ What is mean: ‘landscape factor’? What is difference between ‘landscape factor’ and ‘landscape class’ and ‘landscape phases’?

Author Response

The following changes have been made in response to your comments and suggestions.
(1) The title of the article has been revised.
(2) The survey, research objectives, methods and results were reorganized and the abstract was rewritten.
(3) The introductory section was revised.
(4) More information related to soil occurrence was added to the study area section.
(5) Some errors in terminology were corrected.
(6) I analyzed the information of two soil profiles within a landscape type. And analyzed the connection between the soil-forming conditions associated with that soil type and the landscape factors.
(7) Some formatting, grammar and misspellings were corrected.
(8) Added some citations.

In response to the discussion you raised, here are some of my points.
(1) First, the small landscape area in my results may be due to the high accuracy of the UAV images. Water within the paddy fields may be identified as waters in the land use classification. The ridge between the fields has a certain slope slope direction which may cause the ridge to be divided into different categories from the surrounding fields. This may lead to an excessive number of species in the classification results, resulting in a reduction in the size of the landscape. Second, it may be due to the fact that the area of the soil series may be very small. For example, differences in microtopography may result in small but different types of soils. These soil series may be combined in the next step of soil mapping. Finally, human influence is a very important factor. For example, the construction of roads and the excavation of ponds will fill the soil, which can seriously interfere with the classification of the landscape. In summary, this landscape classification system provides the basis for soil surveys and soil classification. Some landscape categories may be small in area and some will be merged in the next study and some will be retained because they do differ somewhat in attributes.

(2) The landscape factor can be understood as the constituents of the landscape, which is oriented to the soil survey. It can mainly reflect certain soil-forming conditions and is important for soil classification. "landscape class" and "landscape phase" should be I have problems with the expression, they should be "landscape category" and "landscape unit". Landscape unit is a bottom unit, which is produced by superimposing landscape factors. As long as there is a difference of landscape factors is a different landscape unit. A landscape category is composed of multiple landscape units, which all have similar combinations of landscape factors.

Thank you very much for your valuable suggestions!

Round 2

Reviewer 2 Report

I propose add some information about your field studies, e.g. soil sampler collection and soil profile analysis etc.

Some comments are provided in the text (enclosed pdf).

Author Response

The following changes have been made in response to your comments and suggestions.

(1) Data Collection and Data Pre-Processing has been added for the collection and documentation of soil profiles.

(2) Fixed some language errors.